# Experiences Shared by the (Future) Public Health Workforce during the COVD-19 Pandemic in Germany: Results of a Survey on Workload, Work Content, and Related Challenges among Students and Young Professionals

**DOI:** 10.3390/ijerph191811444

**Published:** 2022-09-11

**Authors:** Florian Fischer, Julia Wicherski, Myriam Tobollik, Timothy McCall

**Affiliations:** 1Institute of Public Health, Charité—Universitätsmedizin Berlin, 10117 Berlin, Germany; 2Bavarian Research Center for Digital Health and Social Care, Kempten University of Applied Sciences, 87437 Kempten, Germany; 3Division 5 Research, Federal Institute for Drugs and Medical Devices, 53175 Bonn, Germany; 4Section II 1.5 Environmental Medicine and Health Effects Assessment, German Environment Agency, 14193 Berlin, Germany; 5Medical School OWL, Bielefeld University, 33615 Bielefeld, Germany

**Keywords:** public health, epidemiology, SARS-CoV-2, employment, education, training

## Abstract

Although previous studies have focused on the adverse effects of the COVID-19 pandemic on various professional groups (particularly in the health and nursing care sector), this study aims to close a research gap by assessing perspectives of students and young professionals in epidemiology and public health in Germany in terms of shifts in workload, work content, and related challenges caused by the pandemic. We conducted a cross-sectional survey between mid-February and mid-March 2022. Quantitative data were analyzed via standardized mean differences. Qualitative data based on answers to open-ended questions were analyzed via a qualitative content analysis. Overall, 172 individuals participated in this survey. Results indicate that students felt burdened the most by lack of exchange with other students and lecturers. Study participants employed in public health experienced changes in their employment because they had changes in their work content- and administration, which was accompanied by a high burden due to the workload. Multiple demands that can have an impact on both acquired skills and mental health during the professional qualification phase were mentioned by the participants. Therefore, more in-depth analyses are needed to investigate the impact the pandemic will have on the (future) public health workforce in the long run.

## 1. Introduction

Since the outbreak of the novel coronavirus COVID-19 (coronavirus disease 2019, which is caused by the pathogen SARS-CoV-2) in Wuhan, China in December 2019, the situation has become a global threat [1]. It has been declared as a worldwide pandemic in March 2020 by the World Health Organization [2], leading to containment measures varying over time and region, such as physical distancing (in terms of local, regional, or national shutdowns or lockdowns), hygiene measures, and the halt of all non-essential travel installed by political administrations all over the world [3,4]. While these strategies are aimed at reducing risks of transmitting the infection, they may—particularly lockdown regulations (e.g., staying at home, distance learning at schools and universities)—hinder important means for socializing, finding a partner, and building meaningful relationships, which are of pivotal importance according to personality development theories [3,5,6].

Therefore, the COVID-19 pandemic is not merely a medical issue, but also a behavioral and social phenomenon that must be addressed by public health measures, of which vaccination and adequate treatment only are parts in a complex network of social, economic, and political interventions necessary to cope with the pandemic [7].

Consequences for human health—in terms of direct and indirect health effects—are compiled in different reviews [8,9] also related to mental health issues among healthcare workers [10,11]. However, the effects may differ due to the heterogeneity of population subgroups. Until now, studies have mainly focused on vulnerable populations, such as children, the elderly, or chronically ill people and focused on issues directly related to health and well-being.

Due to the dearth of evidence of COVID-19 pandemic’s impact on the study and working conditions of young professionals in the area of public health and epidemiology, we focused on this subgroup for closer investigation. Previous literature has shown that the COVID-19 pandemic in Germany, as in many other parts of the world, resulted in additional challenges for students enrolled in health degree programs [12]. In addition to general physical distancing measures and the closure of businesses, higher education mainly switched to distance learning only, demanding major restructuring of curricula and examinations. Furthermore, students enrolled in health-related higher education (such as epidemiologic students or public health degree students) were particularly affected by the COVID-19 pandemic as their workforce was especially demanded during this crisis [12,13]. 

Amidst this backdrop, many global and public health projects have been interrupted, with research activities involving travel, study recruitment, and data collection coming to a temporary halt and/or needing to be rethought [14]. The impact of temporarily halting projects and having to rethink them impacts researchers differently. While such repercussions impact senior researchers through funding competition postponements or changes in project timelines [15], additional burdens fall on early-career researchers who may not have secure academic positions and/or funding [14,16,17].

Furthermore, this precariousness in the pandemic context is accentuated by their limited decision-making abilities; strict obligations in meeting deadlines, which are instrumental to personal gains, such as graduation or career advancement; and salary uncertainty, as income is often fueled by larger projects submitted to funding competitions that may have been postponed [14].

For these reasons, we conducted a cross-sectional study among young professionals in public health and epidemiology in Germany to assess their experiences related to workload, work content, and related challenges during the COVID-19 pandemic.

## 2. Materials and Methods

### 2.1. Study Design and Participants

We conducted an online-based survey which was based on standardized items and additionally two open-ended questions to retrospectively assess workload, work content, and related challenges during the COVID-19 pandemic among young professionals in epidemiology and public health. We defined young professionals either as current students in epidemiology- or public-health-related study programs or people who are in their first years of employment within the public health workforce.

Study participants were contacted via the e-mail distribution list and Facebook group section of early-career researchers of the German Society of Epidemiology (DGEpi) and via the e-mail distribution list the German Network of Young Professionals in Public Health (NOEG). In addition, several universities in Germany offering study programs related to public health and epidemiology were asked to forward the invitation via e-mail distribution list among their students. We identified the respective e-mail addresses for the latter through an extraction of relevant study programs from the Higher Education Compass, which is a public database of the German Rectors’ Conference publishing information from state and state-recognized German universities about their study and doctoral programs.

The survey was prepared and released via EvaSys. It was accessible from February 14 to 10 March 2022. We received an ethical waiver for the study from the ethics committee of Bielefeld University (2022–023; 27 January 2022).

### 2.2. Data Collection 

We designed a questionnaire for this study which considered the results of previous studies, which mainly focused on stress levels, mental health, or the appraisal on digital teaching within the university setting. Instead, we assessed data on workload, changes in work content, and challenges related to these issues. The questionnaire contains two socio-demographic questions on age (in years) and gender (male, female, diverse) and six questions about study and working status and conditions (e.g., current number of semesters absolved, graduation status during the COVID-19 pandemic, job type, and job experience). In addition, there were overall 15 either four- or five-tiered Likert scales in which study participants were asked to appraise some statements regarding their own situation. We used a filter beforehand to pose the statements regarding study condition (5 items) and working conditions (10 items) only to those study participants who stated beforehand that they were studying, working, or both.

The five statements for assessing specific study conditions (and one further item for additional conditions) all used a four-tiered Likert scale from “not burdened” to “very burdened”. All statements started with “During the COVID-19 pandemic, I experienced…” and went on with the phrases described in Table 1 to be judged according to their burden on the Likert scale. The questions were taken from a previous study which aimed to assess the study conditions and stress levels of university students during the COVID-19 pandemic in Germany [18].

For assessing working conditions, we used 10 items and one further item to allow respondents adding an additional condition. Here, the answering options at the Likert scales differed according to the question. All statements started with “During the COVID-19 pandemic, I experienced…” and were closed by the phrases described in Table 2. The questions were derived from a study by Gao et al., who described challenges perceived by scientists during the COVID-19 pandemic [19]. 

In addition, the questionnaire included two open-ended questions to assess (1) changes in methods and contents of research projects and (2) potentially facilitating factors for conducting research during the COVID-19 pandemic. 

The questionnaire was pretested by four students. They not only tested the online-based questionnaire for technical functionality, but also provided feedback to questions which were not easy to understand or difficult to answer. Based on their feedback, we adjusted the questionnaire. 

### 2.3. Data Analyses 

The cross-sectional design allows us to quantify standardised differences stratified for participants characteristics. To see who was affected most and in which areas due to the COVID-19 pandemic and the associated containment measures as well as changes in workload and work content for the (future) public health workforce, we quantified standard mean differences (SMD). We conducted SMD analysis for two subsets, namely students and people currently employed within the public health workforce. For students, we considered gender (female, male), age (in years), number of semesters (below 3, 4–10, above 10), and graduation status (yes, no) in the SMD analyses. For people being currently active in the public health workforce, we considered gender (female, male), age (in years), job experience (below 2 years, 2–10 years, above 10 years), and job type (research, education, public health service) in the SMD analyses. The statistical software R version 4.2.0 was used for the analysis.

For qualitative data assessed via the open-ended questions, we used content analysis. We used a deductive approach based on the topics which have been assessed within the quantitative items of the questionnaire for the first step of categorizing the statements. This was followed by an inductive approach to generate further sub-themes based on the content of the statements itself. 

The quantitative and qualitative results are combined in the results section because the qualitative statements elaborated more on specific quantitative results and provided further explanations or concretization. 

## 3. Results

### 3.1. Characteristics of Study Participants

Overall, 172 participants completed the questionnaires. Their ages range from 18 to 61 years, with a median age of 28 years and a mean of 29.71 years. A total of 38 males (22.1%), 133 females (77.3%), and one person of diverse gender (0.6%) answered the questionnaire. From the 172 participants, 116 were students, while 102 were employed in the public health workforce at the time of data collection (Table 3). Combining these two variables gives 53 (30.8%) people working and studying, 63 (36.6%) merely studying, 49 (28.5%) merely working, six (3.5%) neither working or studying, and one person providing no answer on this issue. 

About one third of students stated to be enrolled in no higher than the third semester at the time of data collection, indicating that these students gained their experiences at university only at times during the COVID-19 pandemic. Furthermore, 46 (27.7%) participants graduated within their study program during the COVID-19 pandemic. Of these, 17 gained a bachelor’s degree, 21 a master’s degree, and eight a doctoral degree. Of the people working, 34 reported work experience of less than two years (indicating entry into employment during the COVID-19 pandemic), 30 people had worked for two to five years, and 36 had worked for more than 10 years, whereas two participants provided no information regarding their work experience (Table 3).

For the analysis of differences in the COVID-19 pandemic situation of students and young professionals, we excluded two people: This applies for the person of diverse gender and the person who did not provide any information on job or student status. Therefore, the analysis was run with 170 participants.

### 3.2. Situation of Students during the COVID-19 Pandemic

Table 4 contains the results for experiences of students during the COVID-19 pandemic. The majority of students reported that they were not burdened (66 out of 116) or less burdened (22 out of 116) in financing their studies during the COVID-19 pandemic, while 15 students felt burdened, and another 7 students felt very burdened in this situation. 

In scope of an affected communication during the COVID-19 pandemic, most students felt burdened or very burdened due to a lack of exchange with their lecturer (23 and 42 out of 116, respectively) or fellow students (38 and 58 out of 116, respectively). Regarding the experience of a lack of access to learning materials during the COVID-19 pandemic, 7 students felt very burdened, 31 felt burdened, 33 felt less burdened and 42 felt not burdened, while 3 students gave no response to this question. There is no trend visible regarding the effects of access to undisturbed or quiet workplaces for students during the COVID-19 pandemic. The distribution of the 116 responding students, ordered from very burdened to not burdened, was 33, 24, 26, and 31.

The opportunity to report further experiences related to their student status during the pandemic was only taken by 19 students. Of them, 12 students reported being very burdened. According to the free-text answers, these burdening situations mainly relate to a lack of professional and social exchange. The respondents indicated inadequate balance between private and study life. A doctoral student illustrated: “No more time for the dissertation. Everything else (research, children) is more urgent.” Apart from taking care of children, it was also emphasized that the COVID-19 pandemic led to limited access to compensatory activities (such as sports and cultural events). Furthermore, conversations were mainly focussed only on one topic—namely the COVID-19 pandemic itself—and social isolation occurred because roommates in shared apartments were not present or places where one could meet each other (e.g., canteen) were closed. In addition, a missing focus on the study was described, as it was not easy to motivate oneself. This was particularly highlighted by one respondent with attention deficit hyperactivity disorder (ADHD): “As a person with ADHD, structure is extremely important to me, especially when it is imposed from the outside”. 

Our analysis reveals no relevant gender differences in the burden of experiences of financing student status, communication with lecturers and fellow students, or access to learning materials or quiet places (all SMDs below 0.500; Table 4). 

However, the age in years category reveals differences in the contact with other students, which was a higher burden for younger students (SMD of 0.789), and in the lack of quite workplaces, which did not burden older students (SMD of 0.882). There seems to be only a small difference between age groups regarding experiences of exchange with lecturers and access to learning materials. 

The number of semesters completed and if the students graduated during the pandemic show small to medium differences in terms of SMDs. Students in higher semesters (≥11 semesters) felt more frequently very burdened by a lack of undisturbed or quiet workplaces compared to participants studying ≤3 semesters in total (SMD of 0.650). A tendency with the same direction is shown for the affected’s exchanges with lecturers: Students in higher semesters felt more frequently very burdened due to this situation during the pandemic (SMD of 0.542).

In order to assess the pre-pandemic experience of student status, we excluded participants who completed ≤3 semesters for a sensitivity analysis of changes in students’ situations during the COVID-19 pandemic. Accordingly, 76 out of 116 students were included in this analysis. Results are displayed in Appendix A and indicate that the experience of being a student for at least 4 semesters had small additional influence on the degree of burden due to lack of exchange with lecturers or fellow students and due to lack of access to undisturbed/quiet workplaces. Subgroup analyses of the SMD showed that younger students (≤27 years old) reported higher burden due to lack of communication with lecturers and fellow students compared to older students. Students aged between 28 and 37 years reported more burden due to lack of access to learning materials compared to younger and compared to older students. Younger students (≤27 years old) reported more burden due to lack of access to undisturbed/quiet workplaces than older ones (Appendix A).

### 3.3. Situation of Young Professionals during the COVID-19 Pandemic

Table 5 shows the experiences of young professionals in the public health workforce during the COVID-19 pandemic. The majority of young professionals reported that they were not or rather not involved with infectious diseases on a professional basis for the first time during the COVID-19 pandemic (63 and 10 out of 102 participants, respectively). Only 22 participants reported that they were (rather) involved with infectious diseases in their work for the first time during this situation. Almost two thirds of participants agreed or rather agreed to the statement that the work content has changed during the pandemic (40 and 26 participants, respectively). Within the open-ended questions, some participants stated that more research was conducted related to infection control, particularly in terms of COVID-19, but also on aspects related to Long COVID. One participant also stated that due to high amounts of money spent on COVID-19-related research, some older third-party research grant applications have been “recycled” and complemented by aspects related to COVID-19. Others highlighted that they “have dealt with health-related inequality and their political and social determinants already beforehand”, but COVID-19 has now served as a “case study” or “magnifying glass” for these issues. One respondent coming from politics emphasized that there has been a shift in thematic focuses, because health has suddenly been included in several other political areas, as proposed by the concept of health in all policies. Finally, it has been stated that the COVID-19 pandemic has “led to conducting fast evidence syntheses (e.g., rapid reviews) and products for knowledge translation (e.g., policy briefs, issue briefs)”. 

Many young professionals reported that their workload during the pandemic greatly increased or increased (38 and 31 out of 102, respectively). In 26 individuals, workload did not change. Only one participant reported a reduced workload. Although studies on COVID-19 were visible quite early, these research activities burdened the researchers: “All projects had to be designed and implemented extremely quickly. This built up a tremendous amount of pressure and led to a really large amount of overtime. In fact, I had to work the most over the Christmas vacations (both 20/21 and 21/22).”

Free-text answers clearly indicated that many researchers had to adapt the design of their studies, mostly for qualitative interviews which were conducted via telephone or videoconferencing tools instead of face-to-face. Although some participants stated that these digital tools for data collection led to saving time and money, some also highlighted that approvals for these adaptations of study protocols needed time and that, overall, these changes lead to “massive time delays” within the projects. One participant pointed to the experience that a change from centralized to hybrid and decentralized studies was observable, which also allows easier access to larger sample sizes.

However, the respondents also acknowledged that some target groups were no longer available for empirical research (e.g., particularly vulnerable groups with limited communication ability or mental disabilities). This has led to “considerable impairment of study designs (suspension/severe delay of survey waves in longitudinal designs, drastic increase of loss to follow-up)”. 

The free-text answers also indicate a debate about the pros and cons of digital technologies for qualitative data collection. On one hand, respondents stated that interviews via videoconference tools went very well, particularly because there was no loss of study participants due to non-attendance. On the other hand, the challenge of limited or instable internet access was mentioned. One critical remark regarding methodology overall was as follows: “Qualitative social research—beyond the individual expert interview—is more difficult or almost impossible to conduct, e.g., ethnomethodological observations, focus groups, etc. A shift to quantitative methods does not always make sense from an epistemological point of view. Here, in case of doubt, one should wait before applying a quantitative questionnaire that cannot depict what was originally issued as the research objective.” 

About half of participants gave no answer to the questions regarding their experience with grants/funding and ethics committees during the COVID-19 pandemic (63 and 68 missing, respectively). Of those who gave a response, 28 felt not or less burdened due to insufficient access to grants/funding or lengthy reviews by ethics committees. One study participant stated that the COVID-19 pandemic has led to faster decisions at the ethic committee. 

Although there was proportionally high number of missing values in the data on contact and collaboration with colleagues and external collaborations, about two thirds of persons who responded to this question felt (very) burdened because of a lack of exchanges with their in-house colleagues. Almost the opposite result has been observed for communication and collaboration with external colleagues. This might be due to the fact that many participants stated the benefits of digital technologies for inter-institutional collaborations and cooperation both nationally and internationally. 

In the scope of experiences with access to training or congress participation during the pandemic, 36 out of 102 young professionals agreed that access was easier; a further 30 rather agreed, another 13 rather disagreed, and 10 disagreed on easier access, while 13 participants gave no response. Some participants explicitly stated that visiting congresses—particularly international ones—was much easier due to digital formats. One participant elaborated: “Participation in digital training courses, congresses and (international) networks is much more convenient. This offers an opportunity for all those who have additional private liabilities (care, children, physical limitations...). Maybe it can be considered as a suitable measure for promoting equality.”

In this context, half of participants disagreed or rather disagreed with the statement that quality of trainings and congresses decreased during the COVID-19 pandemic (20 and 32 participants, respectively). Another 13 young professionals rather agreed to decreased quality, and 10 agreed with this situation, while 29 participants gave no answer to this question.

Moreover, only 11 young professionals took the opportunity to report further experiences related to their working situation during the COVID-19 pandemic. Of the 11 individuals, 7 reported very burdened situations, which mainly related to taking care of children (e.g., coordinating home schooling; closed or at least partially restricted availability of kindergartens). 

Of the 102 people who were employed in the public health workforce during the COVID-19 pandemic, gender did not have a relevant influence on the perceived burden assessed in our study. The SMDs displayed in Table 5 vary from 0.234 in the category of quality of training, and conferences decreased to 0.724 in the category regarding access to training or congresses. It seems that females agreed or rather agreed more frequently to the experience of an easier access to training/congress participation compared to males. Moreover, the trend for increased workload is displayed for both genders, but there seems to be a difference in the extent, with larger increases reported by females. 

Participants aged 38 years or older seem to have more often experienced changes in their work content during the pandemic because 22 of 26 of them agreed or rather agreed to this statement compared to 25 of 48 participants aged between 28–37 years and 19 of 28 participants aged 27 years or younger (SMD of 0.570). Similar differences are shown for years of work experience (SMD of 0.590). Furthermore, participants aged 38 years or older seem to disagree or rather disagree more often to the statement of decreased quality of trainings/congresses during the pandemic compared to the younger age groups (SMD of 0.701), but it should be considered that only 50% of the youngest age group provided a response to this question.

Study participants working in the public health service tended to become more often involved with infectious diseases for the first time during the COVID-19 pandemic compared to participants not working in this profession (SMD of 0.685). Small differences (SMD ≤ 0.3) are displayed for the first involvement with infectious diseases between participants working in the research sector or not and participants working in the education sector or not. Work content seems to change more often in educational professions than non-educational professions (SMD of 0.805) and even more in the public health service sector compared to participants not working in the public health service sector (SMD of 0.957).

Easier access to trainings/congresses was reported more often in participants working in research areas compared to non-research areas (SMD of 0.755), whereas the difference for participants working in educational professions versus non-educational professions was smaller (SMD of 0.422). 

The workload of participants in the public health service seems to be more often increased than the workload of participants not working in the research area (SMD of 0.769). For the profession of education, difference in workload seems smaller (SMD of 0.363). The questions regarding grands/funding, colleagues, collaborations, ethics committees, and target groups applied only to participants working in research. Accordingly, no comparison is displayed. Researchers also working in education seem to be more burdened due to insufficient access to grands/funding compared to researchers not also working in education (SMD of 0.685).

## 4. Discussion

Our study shows that the (future) public health workforce in Germany has been burdened due to various aspects during the COVID-19 pandemic not only related to workload but also due to missing social and professional interaction. This applies to participants who were studying or employed at that time. Among public health professionals, one can expect particular challenges in retaining work–life balance during the COVID-19 pandemic [20] because their expertise and workforce was urgently needed [21,22,23]. However, there were also some aspects in which the burden has been perceived quite differently in our study, emphasizing that targeted or tailored approaches are needed. For example, strengthening digital formats both for research and education have either been judged as positive or burdensome. In addition, there have also been serious concerns about quality and validity of research which has been conducted during the COVID-19 pandemic because study designs had to be adjusted to deal with regulations and restrictions caused by the pandemic and its associated containment measures.

To our knowledge, this is the first study focusing on the impact of the COVID-19 pandemic on workload, work content, and related challenges among the public health workforce in Germany which includes data on students and young professionals. Previous studies in the context of the COVID-19 pandemic have either focused on specific aspects—particularly related to digital teaching—within medical students or healthcare personnel [24,25,26] or on mental health issues due to social isolation [1,27]. Although knowledge is available that as an adverse effect of the COVID-19 pandemic education and job training have been disrupted or delayed for many young adults and traditional ways of working encountered serious challenges [28], there has been limited attention on the public health workforce [29,30] in Germany. Previous international studies have already shed some light on exacerbated challenges which have been experienced by the public health workforce, leading to symptoms of posttraumatic stress disorder [31] and burnout [32,33], which may be the results of a shifting focus from normal duties to COVID-19 response activities, higher workload, and job-related threats [31,34]. Therefore, it is imperative to focus much more on the well-being of public health professionals [30,35], which might adversely be impacted due work strains. 

The results of our study indicate not only changes in the work content, but also an overall higher workload. The COVID-19 pandemic has led to an increase in the urgency and scope of the work of public health professionals and reduced their capacity to carry out core services [36]. In particular, young professionals with few years of work experience, who experience such a situation directly at the beginning of their professional life, may be adversely affected. A previous study emphasized that first experiences with such a public health emergency were associated with the intention to leave the job among employees of the public health workforce [29]. Although the public health workforce is an essential part for successfully combating the COVID-19 pandemic [37], one must acknowledge that there is an urgent need to recruit and retain young professionals in this area. To do so, systematic and organizational changes are needed [34,38] to make the public health workforce an attractive area to be employed in.

Until now, the public health workforce had to operate with minimal resources and was characterized by a “lack of continuing education, shortage of professional staff, low wages, and a lack of professional organizations to safeguard employment rights” [39]. During the COVID-19 pandemic, young professionals may have experienced a frustrating disconnect between demand and availability of public health infrastructure—consisting of a capable and qualified workforce, up-to-date data and information systems, and agencies capable of assessing and responding to public health needs—on a daily basis [23]. However, the COVID-19 pandemic has led to a public and political focus on these issues. Therefore, it has created a window of opportunity to create change [40]. This might be particularly useful for young professionals, who have great potential in the field of public health [23].

For attracting these young professionals, digital solutions may serve as valuable tools because they are flexible and affordable [41] and allow easier communication between various actors. However, our study results show divergent experiences of the public health workforce in this regard: Although external cooperation was accelerated due to the digital communication [42], several respondents—both students and young professionals—were burdened due to a lack of exchange with other persons such as colleagues or lecturers. It has already been described in previous literature that the digital communication during the COVID-19 pandemic led to decreases in synchronous communication and increases in asynchronous communication, which make it harder for employees to acquire and share new information [43]. Especially for young people during their study or early professional life, losing a format of exchange and discussion is an aspect that has a negative impact on satisfaction with studies and work and, thus, on their quality of life [44]. Therefore, one must acknowledge the opportunities for digitalization which have also been mentioned by our study participants, such as easier access to training courses, but also must consider their boundaries in avoiding social isolation and overburdening.

A major topic within our analysis has been modifications of research projects. Not only that the scientific rigor might be questioned [45,46] but that adjustment of study protocols or time delays in collecting data have burdened the researchers. This emphasizes the need for supporting young investigators, particularly in times of uncertainty, by increasing connectivity in the field of study, intensifying communication with supervisors and colleagues, broadening mentoring opportunities, and increasing the flexibility of research and funding programs [47].

### Limitations

Several limitations must be considered when interpreting the results. Firstly, we cannot say anything about the representativeness due to the sampling method. We tried to use several distribution lists to gain a large sample size. However, online-based recruiting is based on people’s willingness to participate in research and may, therefore, be biased. This study was planned as an exploratory study and without the aim of multivariable analyses. Therefore, no sample size calculation has been conducted. Secondly, the timing of questions only provides a snapshot in a dynamic pandemic. We asked for a retrospective assessment which covered the time “during the pandemic”. It might be possible that the respective answers would have been different when focusing on specific time points within the COVID-19 pandemic. Thirdly, we did not use validated instruments for data collection but tried to assess the most important points which have been described in studies on other target groups and validated these items in a pretest. Fourthly, there are partly very small numbers in some groups for the SMDs, which hinders validity.

## 5. Conclusions

Overall, the study participants, as part of the (future) public health workforce, expressed multiple demands that can have an impact on both acquired skills and mental health during the professional qualification phase. The analysis indicates that challenges caused by the COVID-19 pandemic and its associated containment measures received divergent appraisals. This illustrates that specific factors may impose different burdens on students and on young professionals, but also within these groups depending on social circumstances and resilience. Therefore, more in-depth analyses are needed to investigate the impact the pandemic will have on the (future) public health workforce in the long run.

## Figures and Tables

**Table 1 ijerph-19-11444-t001:** Items for assessing study conditions.

Variable Label	“During the COVID-19 Pandemic, I Experienced…”	Answer Options
Money	“…financing my student status as…”	“not burdened” to “very burdened”
Lecturer	“…lack of exchange with lecturers as…”	
Co-student	“…lack of exchange with fellow students as…”	
Materials	“…lack of access to learning materials as…”	
Quiet place	“…lack of access to undisturbed/quiet workplaces as…”	
Something else	“…something else I would like mention as…”	

**Table 2 ijerph-19-11444-t002:** Items for assessing working conditions.

Variable Label	“During the COVID-19 Pandemic, I Experienced…”	Answer Options
Infections	“…I became involved in infectious diseases either professionally or during my studies for the first time.”	“disagree” to “agree “
Content	“…my work content has changed.”	“disagree” to “agree”
Workload	“…how has your average workload changed as a result of the COVID-19 pandemic?”	“greatly reduced” to “greatly increased”
Grants	“…I had insufficient access to necessary funding/grants.”	“not burned” to “very burdened”
Ethics	“…I experienced lengthy reviews by the ethic committee.”	“not burned” to “very burdened”
Colleagues	“…I experienced lack of exchange with colleagues from the own institute/department.”	“not burned” to “very burdened”
Consortium	“…I had lack of opportunities for cooperation with other (research) institutes.”	“not burned” to “very burdened”
Target group	“…I had insufficient access to the target group during data collection.”	“not burned” to “very burdened”
Access training	“…access to training of congress participation was easier.”	“disagree” to “agree”
Quality training	“…training/congress quality decreased	“disagree” to “agree”
Something else	“…I experienced something else that I would like to notice here.”	“not burned” to “very burdened”

**Table 3 ijerph-19-11444-t003:** Characteristics of study participants (*n* = 172).

Characteristics	*n*	%
Gender		
Male	38	22.1
Female	133	77.3
Diverse	1	0.6
Age		
<20 years	6	3.5
20–24 years	48	27.9
25–29 years	42	24.4
30–34 years	35	20.3
≥35 years	41	23.8
Studying and employment status		
Student	63	36.6
Student and employed in the public health workforce	53	30.8
Employed in the public health workforce	49	28.5
None or no information	7	4.1
Current semester ^1^		
≤3	37	32.7
4–10	48	42.5
≥11	28	24.8
Graduation during COVID-19 pandemic ^2^		
No graduation	126	73.3
Bachelor’s degree	17	9.9
Master’s degree	21	12.2
Doctoral degree	2	4.7
Job type ^3^		
Research	66	64.7
Education	23	22.5
Public health services	29	28.4
Else (e.g., politics)	22	21.6
Job experience ^4^		
<2 years	34	34.0
2–10 years	51	51.0
>10 years	33	33.0

^1^ This analysis applies only to those students who were enrolled in a study program at the time of data collection (*n* = 63). ^2^ This analysis applies to all participants because it was retrospectively assessed whether one graduated in a study program during the COVID-19 pandemic. ^3^ This analysis applies to all people who stated that they were employed at the time of data collection (*n* = 102). The percentage does not sum to 100% due to the possibility of multiple responses. ^4^ This analysis applies to all people who states to be employed at the time of data collection and who provided information about their job experience (*n* = 100).

**Table 4 ijerph-19-11444-t004:** Experiences of students, stratified for gender, age, number of semesters completed, and graduation status (*n* = 116).

	Gender			Age in Years				Semesters Absolved				Graduation during COVID-19 Pandemic		
	Male	Female	SMD	≤27	28–<38	≥38	SMD	≤3	4–10	≥11	SMD	Yes	No	SMD
*n*	28	87		71	35	10		37	48	28		24	90	
Money			0.455				0.778				0.352			0.315
very burdened	0 (0.0)	6 (7.1)		3 (4.4)	3 (8.8)	1 (12.5)		4 (11.1)	1 (2.2)	2 (7.4)		1 (4.2)	6 (7.1)	
burdened	3 (12.0	12 (14.3)		11 (16.2)	4 (11.8)	0 (0.0)		6 (16.7)	5 (10.9)	4 (14.8)		3 (12.5)	12 (14.3)	
less burdened	4 (16.0)	18 (21.4)		16 (23.5)	6 (17.6)	0 (0.0)		5 (13.9)	13 (28.3)	4 (14.8)		7 (29.2)	14 (16.7)	
not burdened	18 (72.0)	48 (57.1)		38 (55.9)	21 (61.8)	7 (87.5)		21 (58.3)	27 (58.7)	17 (63.0)		13 (54.2)	52 (61.9)	
NA = 6														
Lecturer			0.309				0.153				0.542			0.254
very burdened	6 (23.1)	17 (20.2)		15 (21.1)	7 (21.2)	1 (14.3)		3 (8.6)	12 (25.0)	8 (29.6)		6 (25.0)	17 (20.0)	
burdened	9 (34.6)	33 (39.3)		26 (36.6)	13 (39.4)	3 (42.9)		10 (28.6)	22 (45.8)	10 (37.0)		7 (29.2)	35 (41.2)	
less burdened	5 (19.2)	23 (27.4)		19 (26.8)	8 (24.2)	2 (28.6)		14 (40.0)	9 (18.8)	6 (22.2)		7 (29.2)	21 (24.7)	
not burdened	6 (23.1)	11 (13.1)		11 (15.5)	5 (15.2)	1 (14.3)		8 (22.9)	5 (10.4)	3 (11.1)		4 (16.7)	12 (14.1)	
NA = 5														
Co-student			0.302				0.789				0.314			0.186
very burdened	14 (51.9)	44 (51.8)		43 (60.6)	12 (36.4)	3 (33.3)		14 (38.9)	28 (58.3)	16 (59.3)		13 (54.2)	45 (51.7)	
burdened	7 (25.9)	30 (35.3)		19 (26.8)	17 (51.5)	2 (22.2)		15 (41.7)	14 (29.2)	8 (29.6)		7 (29.2)	30 (34.5)	
less burdened	3 (11.1)	7 (8.2)		6 (8.5)	1 (3.0)	3 (33.3)		3 (8.3)	4 (8.3)	2 (7.4)		3 (12.5)	7 (8.0)	
not burdened	3 (11.1)	4 (4.7)		3 (4.2)	3 (9.1)	1 (11.1)		4 (11.1)	2 (4.2)	1 (3.7)		1 (4.2)	5 (5.7)	
NA = 3														
Materials			0.247				0.397				0.429			0.160
very burdened	3 (11.1)	4 (4.7)		4 (5.6)	3 (9.1)	0 (0.0)		1 (2.8)	4 (8.3)	2 (7.4)		2 (8.3)	5 (5.7)	
burdened	7 (25.9)	23 (27.1)		19 (26.8)	10 (30.3)	2 (22.2)		9 (25.0)	14 (29.2)	8 (29.6)		7 (29.2)	23 (26.4)	
less burdened	7 (25.9)	26 (30.6)		23 (32.4)	7 (21.2)	3 (33.3)		9 (25.0)	18 (37.5)	5 (18.5)		6 (25.0)	27 (31.0)	
not burdened	10 (37.0)	32 (37.6)		25 (35.2)	13 (39.4)	4 (44.4)		17 (47.2)	12 (25.0)	12 (44.4)		9 (37.5)	32 (36.8)	
NA = 3														
Quiet place			0.230				0.822				0.650			0.403
very burdened	8 (28.6)	25 (29.4)		21 (29.6)	12 (35.3)	0 (0.0)		4 (11.1)	16 (33.3)	12 (42.9)		10 (41.7)	23 (26.1)	
burdened	4 (14.3)	19 (22.4)		16 (22.5)	7 (20.6)	1 (11.1)		6 (16.7)	11 (22.9)	7 (25.0)		3 (12.5)	20 (22.7)	
less burdened	7 (25.0)	19 (22.4)		16 (22.5)	6 (17.6)	4 (44.4)		9 (25.0)	11 (22.9)	5 (17.9)		6 (25.0)	20 (22.7)	
not burdened	9 (32.1)	22 (25.9)		18 (25.4)	9 (26.5)	4 (44.4)		17 (47.2)	10 (20.8)	4 (14.3)		5 (20.8)	25 (28.4)	
NA = 2														
Something else			0.761				1.242				1.170			1.386
very burdened	4 (80.0)	8 (57.1)		8 (72.7)	3 (50.0)	1 (50.0)		4 (80.0)	3 (37.5)	4 (80.0)		2 (33.3)	10 (76.9)	
burdened	1 (20.0)	3 (21.4)		2 (18.2)	2 (33.3)	0 (0.0)		0 (0.0)	3 (37.5)	1 (20.0)		2 (33.3)	2 (15.4)	
less burdened	0 (0.0)	1 (7.1)		1 (9.1)	0 (0.0)	0 (0.0)		0 (0.0)	1 (12.5)	0 (0.0)		0 (0.0)	1 (7.7)	
not burdened	0 (0.0)	2 (14.3)		0 (0.0)	1 (16.7)	1 (50.0)		1 (20.0)	1 (12.5)	0 (0.0)		2 (33.3)	0 (0.0)	
NA = 97														

Legend: “During the COVID-19 pandemic, I experienced…” Money: “…financing my student status as…”; Lecturer: “…lack of exchange with lecturers as…”; Co-student: “…lack of exchange with fellow students as…”; Material: “…lack of access to learning materials as…”; Quiet place: “…lack of access to undisturbed/quiet workplaces as…”; Something else: “…something else I would like mention as…”. NA: not applicable; SMD: standard mean difference.

**Table 5 ijerph-19-11444-t005:** Experiences of employed participants, stratified for gender, age, years of work experience, and profession (*n* = 102).

	Gender			Age in Years				Years of Work Experience				Profession: Research			Profession: Education			Profession: Public Health Service		
	Male	Female	SMD	≤27	28–<38	≥38	SMD	≤2	2–<10	≥10	SMD	No	Yes	SMD	No	Yes	SMD	No	Yes	SMD
** *n* **	21	81	28	48	26	34	51	15	36	66	79	23		73	29
**Infections**			0.521				0.402				0.356			0.225			0.331			0.685
agree	4 (21.1)	9 (11.8)		2 (8.0)	8 (17.4)	3 (12.5)		5 (16.7)	6 (12.5)	2 (13.3)		6 (18.2)	7 (11.3)		11 (15.1)	2 (9.1)		5 (7.2)	8 (30.8)	
rather agree	1 (5.3)	8 (10.5)		5 (20.0)	3 (6.5)	1 (4.2)		5 (16.7)	2 (4.2)	2 (13.3)		3 (9.1)	6 (9.7)		8 (11.0)	1 (4.5)		7 (10.1)	2 (7.7)	
rather disagree	4 (21.1)	6 (7.9)		3 (12.0)	4 (8.7)	3 (12.5)		2 (6.7)	6 (12.5)	2 (13.3)		4 (12.1)	6 (9.7)		7 (9.6)	3 (13.6)		9 (13.0)	1 (3.8)	
disagree	10 (52.6)	53 (69.7)		15 (60.0)	31 (67.4)	17 (70.8)		18 (60.0)	34 (70.8)	9 (60.0)		20 (60.6)	43 (69.4)		47 (64.4)	16 (72.7)		48 (69.6)	15 (57.7)	
NA = 7																				
**Content**			0.418				0.570				0.590			0.516			0.805			0.957
agree	8 (40.0)	32 (40.5)		13 (50.0)	16 (34.0)	11 (42.3)		15 (48.4)	20 (39.2)	5 (33.3)		17 (48.6)	23 (35.9)		34 (44.7)	6 (26.1)		21 (29.2)	19 (70.4)	
rather agree	5 (25.0)	21 (26.6)		6 (23.1)	9 (19.1)	11 (42.3)		4 (12.9)	14 (27.5)	7 (46.7)		9 (25.7)	17 (26.6)		14 (18.4)	12 (52.2)		21 (29.2)	5 (18.5)	
rather disagree	1 (5.0)	12 (15.2)		2 (7.7)	10 (21.3)	1 (3.8)		5 (16.1)	6 (11.8)	2 (13.3)		6 (17.1)	7 (10.9)		12 (15.8)	1 (4.3)		12 (16.7)	1 (3.7)	
disagree	6 (30.0)	14 (17.7)		5 (19.2)	12 (25.5)	3 (11.5)		7 (22.6)	11 (21.6)	1 (6.7)		3 (8.6)	17 (26.6)		16 (21.1)	4 (17.4)		18 (25.0)	2 (7.4)	
NA = 3																				
**Workload**			0.516				0.412				0.263			0.715			0.363			0.769
greatly reduced	0 (0.0)	0 (0.0)		0 (0.0)	0 (0.0)	0 (0.0)		0 (0.0)	0 (0.0)	0 (0.0)		0 (0.0)	0 (0.0)		0 (0.0)	0 (0.0)		0 (0.0)	0 (0.0)	
reduced	0 (0.0)	1 (1.3)		0 (0.0)	0 (0.0)	1 (3.8)		0 (0.0)	1 (2.0)	0 (0.0)		0 (0.0)	1 (1.6)		1 (1.4)	0 (0.0)		1 (1.4)	0 (0.0)	
unchanged	6 (28.6)	20 (26.7)		7 (26.9)	14 (31.8)	5 (19.2)		8 (25.8)	15 (30.6)	3 (21.4)		7 (21.2)	19 (30.2)		20 (27.4)	6 (26.1)		22 (31.4)	4 (15.4)	
increased	10 (47.6)	21 (28.0)		10 (38.5)	10 (22.7)	11 (42.3)		10 (32.3)	16 (32.7)	4 (28.6)		6 (18.2)	25 (39.7)		21 (28.8)	10 (43.5)		26 (37.1)	5 (19.2)	
greatly increased	5 (23.8)	33 (44.0)		9 (34.6)	20 (45.5)	9 (34.6)		13 (41.9)	17 (34.7)	7 (50.0)		20 (60.6)	18 (28.6)		31 (42.5)	7 (30.4)		21 (30.0)	17 (65.4)	
NA = 6																				
**Grants**			0.429				0.621				0.901			NaN			0.685			1.318
very burdened	1 (10.0)	5 (17.2)		1 (20.0)	3 (13.6)	2 (16.7)		2 (22.2)	3 (12.5)	1 (20.0)		0 (NaN)	6 (15.4)		2 (8.3)	4 (26.7)		6 (16.2)	0 (0.0)	
burdened	1 (10.0)	4 (13.8)		1 (20.0)	2 (9.1)	2 (16.7)		1 (11.1)	2 (8.3)	2 (40.0)		0 (NaN)	5 (12.8)		2 (8.3)	3 (20.0)		4 (10.8)	1 (50.0)	
less burdened	3 (30.0)	4 (13.8)		0 (0.0)	6 (27.3)	1 (8.3)		0 (0.0)	6 (25.0)	0 (0.0)		0 (NaN)	7 (17.9)		5 (20.8)	2 (13.3)		7 (18.9)	0 (0.0)	
not burdened	5 (50.0)	16 (55.2)		3 (60.0)	11 (50.0)	7 (58.3)		6 (66.7)	13 (54.2)	2 (40.0)		0 (NaN)	21 (53.8)		15 (62.5)	6 (40.0)		20 (54.1)	1 (50.0)	
NA = 63																				
**Ethics**			0.420				0.856				1.276			NaN			0.525			1.268
very burdened	1 (11.1)	1 (4.0)		0 (0.0)	1 (4.5)	1 (10.0)		0 (0.0)	1 (4.5)	1 (20.0)		0 (NaN)	2 (5.9)		0 (0.0)	2 (11.8)		2 (6.2)	0 (0.0)	
burdened	1 (11.1)	3 (12.0)		0 (0.0)	3 (13.6)	1 (10.0)		0 (0.0)	2 (9.1)	2 (40.0)		0 (NaN)	4 (11.8)		2 (11.8)	2 (11.8)		3 (9.4)	1 (50.0)	
less burdened	1 (11.1)	6 (24.0)		0 (0.0)	5 (22.7)	2 (20.0)		1 (16.7)	6 (27.3)	0 (0.0)		0 (NaN)	7 (20.6)		4 (23.5)	3 (17.6)		7 (21.9)	0 (0.0)	
not burdened	6 (66.7)	15 (60.0)		2 (100.0)	13 (59.1)	6 (60.0)		5 (83.3)	13 (59.1)	2 (40.0)		0 (NaN)	21 (61.8)		11 (64.7)	10 (58.8)		20 (62.5)	1 (50.0)	
NA = 68																				
**Colleagues**			0.611				0.586				0.918			NaN			0.404			0.559
very burdened	2 (12.5)	16 (33.3)		8 (44.4)	5 (17.9)	5 (27.8)		7 (35.0)	11 (30.6)	0 (0.0)		0 (NaN)	18 (28.1)		14 (33.3)	4 (18.2)		16 (27.1)	2 (40.0)	
burdened	9 (56.2)	16 (33.3)		6 (33.3)	11 (39.3)	8 (44.4)		6 (30.0)	14 (38.9)	5 (71.4)		0 (NaN)	25 (39.1)		14 (33.3)	11 (50.0)		24 (40.7)	1 (20.0)	
less burdened	3 (18.8)	12 (25.0)		4 (22.2)	7 (25.0)	4 (22.2)		5 (25.0)	7 (19.4)	2 (28.6)		0 (NaN)	15 (23.4)		10 (23.8)	5 (22.7)		14 (23.7)	1 (20.0)	
not burdened	2 (12.5)	4 (8.3)		0 (0.0)	5 (17.9)	1 (5.6)		2 (10.0)	4 (11.1)	0 (0.0)		0 (NaN)	6 (9.4)		4 (9.5)	2 (9.1)		5 (8.5)	1 (20.0)	
NA = 38																				
**Consortium**			0.384				0.737				0.697			NaN			0.514			0.712
very burdened	2 (13.3)	3 (8.3)		1 (12.5)	1 (3.8)	3 (17.6)		1 (8.3)	3 (9.7)	1 (14.3)		0 (NaN)	5 (9.8)		2 (6.7)	3 (14.3)		5 (10.6)	0 (0.0)	
burdened	4 (26.7)	9 (25.0)		3 (37.5)	8 (30.8)	2 (11.8)		4 (33.3)	8 (25.8)	1 (14.3)		0 (NaN)	13 (25.5)		10 (33.3)	3 (14.3)		11 (23.4)	2 (50.0)	
less burdened	6 (40.0)	11 (30.6)		1 (12.5)	8 (30.8)	8 (47.1)		2 (16.7)	10 (32.3)	4 (57.1)		0 (NaN)	17 (33.3)		10 (33.3)	7 (33.3)		16 (34.0)	1 (25.0)	
not burdened	3 (20.0)	13 (36.1)		3 (37.5)	9 (34.6)	4 (23.5)		5 (41.7)	10 (32.3)	1 (14.3)		0 (NaN)	16 (31.4)		8 (26.7)	8 (38.1)		15 (31.9)	1 (25.0)	
NA = 51																				
**Target group**			0.465				0.671				0.485			NaN			0.662			1.268
very burdened	4 (33.3)	7 (19.4)		1 (9.1)	4 (17.4)	6 (42.9)		2 (14.3)	3 (18.8)	6 (35.3)		0 (NaN)	11 (22.9)		5 (16.7)	6 (33.3)		11 (23.9)	0 (0.0)	
burdened	4 (33.3)	13 (36.1)		6 (54.5)	7 (30.4)	4 (28.6)		6 (42.9)	5 (31.2)	5 (29.4)		0 (NaN)	17 (35.4)		13 (43.3)	4 (22.2)		16 (34.8)	1 (50.0)	
less burdened	3 (25.0)	8 (22.2)		2 (18.2)	7 (30.4)	2 (14.3)		3 (21.4)	6 (37.5)	2 (11.8)		0 (NaN)	11 (22.9)		8 (26.7)	3 (16.7)		10 (21.7)	1 (50.0)	
not burdened	1 (8.3)	8 (22.2)		2 (18.2)	5 (21.7)	2 (14.3)		3 (21.4)	2 (12.5)	4 (23.5)		0 (NaN)	9 (18.8)		4 (13.3)	5 (27.8)		9 (19.6)	0 (0.0)	
NA = 54																				
**Access training**			0.724				0.068				0.394			0.755			0.422			0.733
agree	6 (31.6)	30 (42.9)		8 (40.0)	18 (41.9)	10 (38.5)		13 (48.1)	18 (38.3)	5 (38.5)		7 (22.6)	29 (50.0)		25 (37.3)	11 (50.0)		30 (46.9)	6 (24.0)	
rather agree	10 (52.6)	20 (28.6)		7 (35.0)	14 (32.6)	9 (34.6)		9 (33.3)	16 (34.0)	4 (30.8)		10 (32.3)	20 (34.5)		22 (32.8)	8 (36.4)		23 (35.9)	7 (28.0)	
rather disagree	3 (15.8)	10 (14.3)		3 (15.0)	6 (14.0)	4 (15.4)		4 (14.8)	5 (10.6)	3 (23.1)		8 (25.8)	5 (8.6)		11 (16.4)	2 (9.1)		7 (10.9)	6 (24.0)	
disagree	0 (0.0)	10 (14.3)		2 (10.0)	5 (11.6)	3 (11.5)		1 (3.7)	8 (17.0)	1 (7.7)		6 (19.4)	4 (6.9)		9 (13.4)	1 (4.5)		4 (6.2)	6 (24.0)	
NA = 13																				
**Quality training**			0.234				0.701				0.575			0.750			0.598			0.750
agree	1 (5.9)	7 (12.5)		4 (28.6)	4 (11.1)	0 (0.0)		4 (19.0)	4 (10.5)	0 (0.0)		7 (24.1)	1 (2.3)		8 (14.5)	0 (0.0)		2 (3.9)	6 (27.3)	
rather agree	3 (17.6)	10 (17.9)		3 (21.4)	6 (16.7)	4 (17.4)		3 (14.3)	6 (15.8)	4 (33.3)		6 (20.7)	7 (15.9)		10 (18.2)	3 (16.7)		9 (17.6)	4 (18.2)	
rather disagree	8 (47.1)	24 (42.9)		4 (28.6)	18 (50.0)	10 (43.5)		9 (42.9)	17 (44.7)	4 (33.3)		9 (31.0)	23 (52.3)		23 (41.8)	9 (50.0)		26 (51.0)	6 (27.3)	
disagree	5 (29.4)	15 (26.8)		3 (21.4)	8 (22.2)	9 (39.1)		5 (23.8)	11 (28.9)	4 (33.3)		7 (24.1)	13 (29.5)		14 (25.5)	6 (33.3)		14 (27.5)	6 (27.3)	
NA = 29																				
**Something else**			3.000				1.054				2.279			NaN			0.967			0.926
very burdened	0 (0.0)	7 (77.8)		1 (50.0)	4 (80.0)	2 (50.0)		1 (33.3)	6 (85.7)	0 (0.0)		0 (NaN)	7 (63.6)		5 (71.4)	2 (50.0)		7 (70.0)	0 (0.0)	
burdened	1 (50.0)	2 (22.2)		1 (50.0)	0 (0.0)	2 (50.0)		1 (33.3)	1 (14.3)	1 (100.0)		0 (NaN)	3 (27.3)		1 (14.3)	2 (50.0)		3 (30.0)	0 (0.0)	
less burdened	0 (0.0)	0 (0.0)		0 (0.0)	0 (0.0)	0 (0.0)		0 (0.0)	0 (0.0)	0 (0.0)		0 (0.0)	0 (0.0)		0 (0.0)	0 (0.0)		0 (0.0)	0 (0.0)	
not burdened	1 (50.0)	0 (0.0)		0 (0.0)	1 (20.0)	0 (0.0)		1 (33.3)	0 (0.0)	0 (0.0)		0 (NaN)	1 (9.1)		1 (14.3)	0 (0.0)		0 (0.0)	1 (100.0)	
NA = 91																				

*Legend:* “During the COVID-19 pandemic…” Infections: “…I became involved in infectious diseases either professionally or during my studies for the first time.”; Content: “…my work content has changed.”; Workload: “…how has your average workload changed as a result of the COVID-19 pandemic?”; Grants: “…I had insufficient access to necessary funding/grants.”; Ethics: “…I experienced lengthy reviews by the ethic committee.”; Colleagues: “…I experienced lack of exchange with colleagues from the own institute/department.”; Consortium: “…I had lack of opportunities for cooperation with other (research) institutes.”; Target group: “…I had insufficient access to the target group during data collection.”; Access: “…access to training of congress participation was easier.”; Quality: “…training/congress quality decreased.”; Something else: “…I experiences something else that I would like to notice here.” NA: not applicable; SMD: standard mean differences.

## Data Availability

Data are available from the corresponding author upon reasonable request.

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
