# Peer review of "Experiences Shared by the (Future) Public Health Workforce during the COVD-19 Pandemic in Germany: Results of a Survey on Workload, Work Content, and Related Challenges among Students and Young Professionals"

_ijerph, 2022, doi:10.3390/ijerph191811444_

Round 1

Reviewer 1 Report

The authors did a great job. The article is fascinating and I could spend many positive words on it. However, I am wondering about the reliability of the measures developed for this study. There is no information provided and I would like to ask the authors to check the reliability of their measures before proceeding with publication. Also, I think it would be more useful for the reader to have the items of the measures separated by the text. This could be done by making a table with items and quoting it in the text. Or, this could be done by putting the items in the appendix. I leave the authors to choose which option works better for them.

Author Response

The authors did a great job. The article is fascinating and I could spend many positive words on it.

Thank you very much for this very positive feedback.

However, I am wondering about the reliability of the measures developed for this study. There is no information provided and I would like to ask the authors to check the reliability of their measures before proceeding with publication.

We have added references to the studies where the items have been taken or derived from.

Also, I think it would be more useful for the reader to have the items of the measures separated by the text. This could be done by making a table with items and quoting it in the text. Or, this could be done by putting the items in the appendix. I leave the authors to choose which option works better for them.

The items have now been displayed in tables as suggested by the reviewer.

Reviewer 2 Report

The title accurately reflects the content of the manuscript, the keywords are well defined, and the abstract fully summarizes the content of the manuscript. The introduction clearly defines the purpose of the manuscript, as well as its relevance. Individual sections of the manuscript are proportional in size, and the authors have consulted current and relevant literature. The research methodology is described in detail, as are all of the data needed for this empirical study. The research results have been presented in a detailed and precise manner, are consistent with the research problem, and correspond to the chosen methodology.
The discussion includes a critical evaluation of the findings, and the conclusions are based on the authors' findings. Although the results are largely predictable, they are valuable for society, practice, and research, and the manuscript as a whole contains elements of scientific originality.

Author Response

The title accurately reflects the content of the manuscript, the keywords are well defined, and the abstract fully summarizes the content of the manuscript. The introduction clearly defines the purpose of the manuscript, as well as its relevance. Individual sections of the manuscript are proportional in size, and the authors have consulted current and relevant literature. The research methodology is described in detail, as are all of the data needed for this empirical study. The research results have been presented in a detailed and precise manner, are consistent with the research problem, and correspond to the chosen methodology.
The discussion includes a critical evaluation of the findings, and the conclusions are based on the authors' findings. Although the results are largely predictable, they are valuable for society, practice, and research, and the manuscript as a whole contains elements of scientific originality.

Thank you very much for your positive feedback on our manuscript.

Reviewer 3 Report

The topic chosen by the authors may be relevant and important for the journal's readership, but the methodological part of the paper is very weak. The authors acknowledge this and list these weaknesses (sample size,  instruments) in the limitation section. I would argue that the article should not be published if these methodological aspects are not met.

Author Response

The topic chosen by the authors may be relevant and important for the journal's readership, but the methodological part of the paper is very weak. The authors acknowledge this and list these weaknesses (sample size, instruments) in the limitation section. I would argue that the article should not be published if these methodological aspects are not met.

Thank you for your feedback. As you have already mentioned, we described the limitations. In general, we aimed to study a sub-population which is quite small and hard to reach during times of the pandemic. For that reason, we are satisfied with the sample size, which allows for descriptive statistics. We never aimed to conduct multivariable analyses which would need a larger sample size and power. From our point of view, this study provides first insights into a group which has not been investigated during the pandemic in Germany beforehand.

To overcome the challenges regarding instrument, we described in more detail where the items have been taken from. These items refer to previous studies which have been used among other population subgroups.

Round 2

Reviewer 3 Report

Thank you for your additions. This article fulfils all the basic requirements of  article. In the future, I would like to see a larger sample size and more advanced statistical analyses.